# Antioxidant Use after Diagnosis of Head and Neck Squamous Cell Carcinoma (HNSCC): A Systematic Review of Application during Radiotherapy and in Second Primary Cancer Prevention

**DOI:** 10.3390/antiox12091753

**Published:** 2023-09-12

**Authors:** Piero Giuseppe Meliante, Carla Petrella, Marco Fiore, Antonio Minni, Christian Barbato

**Affiliations:** 1Department of Sense Organs DOS, Sapienza University of Rome, Viale del Policlinico 155, 00161 Roma, Italy; pierogiuseppe.meliante@uniroma1.it; 2Institute of Biochemistry and Cell Biology (IBBC), National Research Council (CNR), Department of Sense Organs DOS, Sapienza University of Rome, Viale del Policlinico 155, 00161 Roma, Italy; carla.petrella@cnr.it (C.P.); marco.fiore@cnr.it (M.F.); 3Division of Otolaryngology-Head and Neck Surgery, Ospedale San Camillo de Lellis, ASL Rieti-Sapienza University, Viale Kennedy, 02100 Rieti, Italy

**Keywords:** second primary cancers, antioxidant, HNSCC, clinical trials

## Abstract

Approximately 5–20% of HNSCC patients experience second primary cancers within the first 5 years of treatment, contributing to high mortality rates. Epidemiological evidence has linked a low dietary intake of antioxidants to an increased risk of cancer, especially squamous cell carcinoma, prompting research into their potential in neoplasm chemoprevention. Cigarette smoking is the primary risk factor for HNSCC, and a diet rich in antioxidants offers protective effects against head and neck cancer. Paradoxically, smokers, who are at the highest risk, tend to consume fewer antioxidant-rich fruits and vegetables. This has led to the hypothesis that integrating antioxidants into the diet could play a role in both primary and secondary prevention for at-risk individuals. Furthermore, some HNSCC patients use antioxidant supplements during chemotherapy or radiotherapy to manage side effects, but their impact on cancer outcomes remains uncertain. This systematic review explores the evidence for the potential use of antioxidants in preventing second primary cancers in HNSCC patients. In conclusion, none of the antioxidants tested so far (α-tocopherol, β-carotene, JP, Isotretinoin, interferon α-2a, vitamin E, retinyl palmitate, N-acetylcysteine) was effective in preventing second primary tumors in HNSCC patients, and they could only be used in reducing the side effects of radiotherapy. Further research is needed to better understand the interplay between antioxidants and cancer outcomes in this context.

## 1. Introduction

Head and neck cancer is a global health challenge, ranking as the sixth most prevalent malignancy worldwide, with an incidence of 650,000 new cases and 350,000 deaths per year [1]. Head and neck squamous cell carcinoma (HNSCC) is the most common histotype. It is a challenging condition to treat, with a 5-year overall survival rate of around 60% [2]. A ‘second primary cancer’ is a separate neoplasm that develops independently from a previous or concurrent cancer in the same individual. The emergence of a second primary cancer poses a significant obstacle to successful long-term outcomes. Within the first 5 years after treatment, approximately 5 to 20% of HNSCC patients develop a second primary cancer [3,4,5], which has been identified as a leading cause of mortality, even in cases of early-stage neoplasms that have been effectively treated [6]. The risk of developing a second primary cancer has been estimated to be 3 to 5% per year from the diagnosis of the primary neoplasm [7].

Pro-oxidant substances or free radicals are molecules equipped with unpaired electrons that, due to their instability, interact with other molecules. Reactive oxygen species (ROS) attack cellular molecules such as fats, proteins, and nucleic acids [8]. Oxidative stress and ROS play causal roles in the development of cancer. ROS damage DNA causing mutations and activating the expression of genes involved in DNA repair and cell proliferation. Antioxidants are molecules able to inhibit oxidation [9]. Epidemiologic evidence has linked low dietary intake of antioxidants to an increased risk of cancer, particularly squamous cell carcinoma [10,11]. Therefore, antioxidants have been studied for their possible use in neoplasm chemoprevention [12]. The primary risk factors for developing HNSCC are cigarette smoking and alcohol consumption [13]. The spread of the human papillomavirus is increasing the incidence of related cervico-cephalic neoplasms [14]. Conversely, a diet rich in antioxidants is a protective factor against the development of head and neck cancer. Patients most at risk, i.e., smokers, are also those who consume the least amount of fruit and vegetables [15,16,17,18,19,20]. This has led to the hypothesis that integrating antioxidants into the diet could not only play a role in primary prevention but also in secondary prevention for at-risk individuals. Furthermore, some HNSCC patients also take antioxidant supplements during chemotherapy or radiotherapy to alleviate the side effects, particularly oral mucositis [21]. However, their impact on cancer outcomes remains uncertain [22,23,24,25]. Given the protective role of antioxidants against tumors, several researchers have independently wondered whether supplementation in patients with HNSCC could reduce the incidence of second primary neoplasms. The aim of this systematic review of the literature is to summarize the evidence and provide an answer on the potential action of these molecules in preventing second primary cancers.

## 2. Materials and Methods

PubMed, Google Scholar, and Scopus databases were interrogated for articles regarding chemoprevention using antioxidants for the development of second primary malignancies in patients with HNSCC. The type of publication taken into consideration for the purposes of this review were only randomized and not randomized clinical trials. We exclusively considered manuscripts published between 1993 and 2023. The search queries used were “head and neck cancer antioxidant prevention; head and neck cancer antioxidant second primary; head and neck cancer antioxidant chemoprevention; head and neck squamous cell carcinoma antioxidant chemoprevention”. The search was restricted to English-language articles only. In vitro and animal-based trials were excluded. After collecting the articles, the duplicates were initially removed through the Mendeley platform, which was also used for the management of the bibliography. After the removal of duplicates, the authors collectively screened the publications by title and abstract. The remaining manuscripts were selected by analyzing the entire text. In this phase, all the articles that did not concern trials on human beings were excluded, as well as all the trials that did not have among the outcomes the incidence of secondary primary carcinomas. Finally, the articles selected for review are summarized in this manuscript and in Table 1. The analysis of the literature, also as regards the final review, contributed to the drafting of the observations contained in the discussion. The critical issues regarding the analyzed protocols were discussed by our team and summarized in the discussion of this article. Furthermore, the discussion was integrated with a brief dissertation on antioxidant molecules which, although they have known efficacy in primary prevention, have not yet been studied for secondary prevention in patients with HNSCC.

The present review was conducted following the 2020 PIRSMA statement. The PRISMA checklist was successfully completed after manuscript composition. Our protocol was registered in the PROSPERO International prospective register of systematic reviews (CRD42023453500).

## 3. Results

### 3.1. Systematic Literature Research

The online database search yielded 201 articles, which were imported into the Mendeley Reference Manager platform and made available to all authors. After nine duplicates were removed, there were 189 papers remaining. The authors screened by title and abstract and excluded 115 manuscripts not addressing the use of antioxidants for HNSCC prevention in human models. For the remaining 76 articles, the authors studied the full text, and 11 trials concerning chemoprevention of secondary malignancies in patients with HSCC were selected (Figure 1).

### 3.2. Clinical Trials and Antioxidants

Bairati et al. conducted a multicentric double-blinded placebo-controlled randomized trial comparing supplementation with α-tocopherol (400 IU/day) and β-carotene (30 mg/day) with placebo in patients with stage I or II HNSCC treated with radiotherapy (RT). Antioxidants were administered from the first day of RT and continued for 3 years after its completion. Among 540 patients enrolled, 273 were assigned to the experimental arm and 267 to the control group. The median followup was 52 months. The main outcome was to assess whether the supplementation would reduce the incidence of second primary cancers, the second objective was the effect of these substances on RT adverse effects, and the third one was the cancer-free survival impact [26]. β-carotene supplementation was discontinued due to the emerging evidence about its possible association with lung cancer development, and the trial was continued with α–tocopherol only [27]. The analysis was also conducted separately for the first 156 patients who used both substances and for the 384 patients subsequently enrolled. The antioxidant group had a statistically significantly higher incidence of second primary cancers than the placebo group (60/1000 person-years vs. 25/1000 person-years, hazard ratio (HR) = 2.42 (95% CI = 1.45 to 4.04), especially when α-tocopherol was the only supplement (HR = 2.88, 95% CI = 1.56 to 5.31). After supplement suspension, however, the experimental group had a lower incidence of second neoplasms (39/1000 people per year against 69/1000 people per year, HR = 0.57 95% CI = 0.31 to 1.0), with even greater significance when α -tocopherol was the only supplement (HR = 0.41, 95% CI = 0.16 to 1.03) [26] (Table 1).

The same authors analyzed the population of this study in relation to cigarette smoking, and they investigated whether the association between smoking and supplementation affected second primary tumors. In total, 60% were smokers before RT (343 of 540), which decreased to 33% at the start of treatment. The increased incidence of outcomes associated with supplementation was observed only among cigarette smokers: HR 2.41 (95% CI: 1.25–4.64) for recurrence, HR 2.26 (95% CI: 1.25–4.64) for all-cause mortality, and HR 5 3.38 (95% CI: 1.11–10.34) for deaths due to the initial HNC. All corresponding HRs among never-smokers were close to 1. The use of α-tocopherol and β-carotene or α-tocopherol alone led to comparable results. Therefore, the authors concluded that the adverse effects observed in their trial were attributable to cigarette smoking [28] (Table 1).

β-carotene has been tested in several other secondary chemoprevention trials. Starting from the assumption that it can reverse precancerous lesions of the oral cavity, its use has been tested to prevent the onset of second primary tumors or local recurrences. A population of 264 patients was randomly divided into a group treated with 50 mg/day of β-carotene or with a placebo. The mean follow-up was 51 months. The authors observed no statistically significant differences between the two groups. The relative risk (RR) of second primary tumors was 0.90 (95% confidence interval (CI), 0.56–1.45). Mortality was not affected (RR, 0.86; 95% CI, 0.52–1.42). However, they observed a possible increased incidence of lung malignancies in the experimental group [29] (Table 1).

A randomized, double-blind, placebo-controlled trial was performed in a population of 134 patients to test the efficacy of JP (NSA International, Memphis, TN, USA), a fruit and vegetable concentrate. Patients of the experimental group (72) showed a significant increase in serum α-carotene (*p* = 0.004), β-carotene (*p* < 0.0001), lutein (*p* = 0.0004), retinol (*p* = 0.045), and α-tocopherol (*p* = 0.023) and a decrease in serum γ-tocopherol (*p* = 0.04) after 3 months. In contrast, the placebo group showed a significant change only of lycopene (*p* = 0.019). Surrogate p27 and Ki-67 outcomes were not significantly affected by the supplement. The incidence of second malignancies, although slightly lower in the JP-treated group, did not have a statistically significant difference (HR = 0.56; 95% CI = 0.20 to 1.57; *p* = 0.27), even after adjusting for patient characteristics (HR = 0.88; 95% CI = 0.25 to 3.11) [30] (Table 1).

**Table 1 antioxidants-12-01753-t001:** Summary of the trials regarding chemoprevention of second primary cancer after head and neck squamous cell carcinoma using antioxidants.

Authors	Year	PMID	Population	Comparison	Results	Notes	Strength of Evidence OCEBM *	Reference
Bairati et al.	2005	15812073	540 patients treated with RT	α-tocopherol (400 IU/day) and β-carotene (30 mg/day) or placebo on the first day of RT and continued for 3 years after the end of RT	Antioxidant group had higher incidence of second primary cancers.		2	[26]
Meyer et al.	2008	18059031	540 patients treated with RT	α-tocopherol (400 IU/day) and β-carotene (30 mg/day) or placebo on the first day of RT and continued for 3 years after the end of RT considering smoking	Smoking did not modify the effects of the supplementation. Cigarette smoking increased risk of second primary tumors.		2	[28]
Mayne et al.	2001	11245451	240 patients	β-carotene (50 mg/day) vs. placebo	No significant effect on second head and neck cancer	Potential increased risk of lung cancer	2	[29]
Datta et al.	2018	28102098	134 patients	JP vs. placebo	No statistically significant effect on second head and neck cancer		2	[30]
Seixas-Silva et al.	2018	15837897	45 patients	Isotretinoin, interferon α-2a, and vitamin E	The bioadjuvant combination is highly effective in preventing recurrence and second primary tumors	No control group	3	[31]
Shin et al.	2001	v11408495	45 patients	Interferon-α, 13-cis-retinoic acid, and α-tocopherol.	The bioadjuvant combination is highly effective in preventing recurrence and second primary tumors	No control group	3	[32]
van Zandwijk et al.	2000	10861309	2592 patients (60% with head and neck cancer and 40% with lung cancer)	(1) retinyl palmitate (300,000 IU/day for 1 year followed by 150,000 IU for successive year), (2) N-acetylcysteine (600 mg/day for 2 years), (3) both compounds, and (4) no intervention.	No difference was seen. Lower incidence of second primary tumors in the no intervention group, not statistically significant.		2	[33,34]
Jyothirmayi et al.	1996	9039219	49 HNSCC treated with Vitamin A and 42 placebo completed 3-year followup	retinyl palmitate (200,000 IU per week for 1 year) vs. placebo	2 recurrences in the placebo group, 0 in the experimental one		2	[35]
Toma et al.	2003	14534715	214 HNSCC stage I-II radically treated (110 experimental vs. 104 placebo)	β-carotene 75 mg/day for 3-month cycles within one month intercycle intervals for a total of 3 years	No statistically significant effect on second head and neck cancer		2	[36]
Khuri et al.	2006	16595780	1190 stage I or II HNSCC patients	Isotretinoin (13-cis-retinoic acid) (30 mg/day) vs. placebo for 3 years	No statistically significant effect on second head and neck cancer			[7]

* Strength of evidence has been estimated following the OCEBM, Oxford Centre for Evidence-Based Medicine [37].

Isotretinoin, interferon α-2a, and vitamin E were used in a phase 2 trial of 45 patients with HNSCC. The authors observed only one case of recurrence and concluded that the bioadjuvant combination is effective in preventing recurrence and second primary tumors [31] (Table 1). In our opinion, the absence of a control group and the small sample size make this conclusion unsatisfactory.

Another phase 2 study investigated the association of interferon-α, 13-cis-retinoic acid, and α-tocopherol at doses of 50 mg/m^2^/d, orally, daily, 3 × 10^6^ IU/m^2^, subcutaneous injection, 3/week, and 1200 IU/d, orally, daily, respectively, for 12 months in locally advanced HNSCC as an adjuvant to therapy. In this case, only one second primary cancer was observed, but no control group was given the phase 2 design [32]. As in the previous study, the use of a control group is necessary (Table 1).

The EUROSCAN study recruited 2592 patients affected by HNSCC (60%) or lung cancer (40%) and evaluated the chemopreventive action of vitamin A (retinyl palmitate) and N-acetylcysteine. The patients were divided into four experimental groups, one receiving retinyl palmitate (300,000 IU/day for 1 year followed by 150,000 IU/day for the subsequent year), a second group receiving N-acetylcysteine (600 mg/day for 2 years), a third receiving both, and a fourth group receiving neither. The authors observed no benefit on the development of second primary neoplasms from this integration; the fourth group that did not receive any treatment had a lower incidence of secondary neoplasms that was not statistically significant [33,34] (Table 1).

The efficacy of vitamin A alone in preventing primary seconds was also one of the outcomes of a randomized trial in which 50 patients were treated with retinyl palmitate (200,000 IU/week for 1 year) and 43 with a placebo. In total, 49 subjects in the experimental group and 42 in the placebo group completed the 3-year followup. The group treated with vitamin A showed no second primary tumors, but a higher incidence of recurrences, while the placebo group showed two cases of second neoplasms but a lower incidence of recurrences [35] (Table 1).

An Italian study evaluated the efficacy of β-carotene on survival, disease-free survival, and the incidence of second primary tumors in patients radically treated for stage I or II HNSCC. A population of 214 patients was randomly divided into a group treated with β-carotene 75 mg/day for 3-month cycles within one-month intercycle intervals for a total of 3 years. After a median followup of 59 months and with 3-year compliance of 68.7% in the experimental group, the authors observed no statistically significant differences for 10-year disease-free survival (75.7% vs. 74.3%, *p* = 0.56) and second primitive neoplasms (RR = 0.99; 95% CI 0.28–3.44) [36] (Table 1).

Isotretinoin (13-cis-retinoic acid), a synthetic vitamin A derivative or retinoid, has been tested at low doses (30 mg/day) for 3 years against a placebo in stage I or II HNSCC patients. In total, 1190 patients were randomly assigned to the experimental or control group. Isotretinoin did not statistically significantly reduce the second primary cancers’ incidence hazard ratio (HR = 1.06, 95%; CI = 0.83 to 1.35) [7] (Table 1).

## 4. Discussion

### 4.1. Results

Of all the studies highlighted, none supported the utility of antioxidants to prevent second primary tumors in patients with HNSCC [26,28,29,30,33,34,35]. Indeed, in some cases, these have proved harmful, albeit not statistically significant [28,35,36]. The only two trials that claim a potential benefit are also the only ones that lack a control group [31,32].

Considering the results highlighted in this review, it is possible to argue that supplements cannot replicate the synergistic effect of the numerous components contained in natural foods. Also, the studies performed on fruit and vegetable concentrates have not shown any benefit from supplementation with antioxidants on the development of secondary neoplasms [30].

Although all the molecules investigated up to now have not proved effective in reducing the incidence of secondary primary neoplasms after diagnosis of HNSCC, our results are not generalizable to all antioxidants. It is imperative to delineate the confines within which our conclusions should be interpreted. The molecules that came under the lens of scrutiny within the purview of our review may not encapsulate the entire gamut of antioxidant possibilities. The antioxidants examined represent a subset, rather than a comprehensive exploration of all possible variants. Although they do not collectively exhibit unequivocal efficacy in reducing secondary primary neoplasms post-HNSCC diagnosis, this does not exclude other possibilities.

Furthermore, a thorough assessment of the molecules already studied demands an exploration of the dosages at which they were administered in the experimental trials. Some molecules might not have exhibited their potential effects at the doses tested, leaving open the possibility of efficacy at different dosage levels. Given the paucity of experiments conducted on the prevention of secondary primary neoplasms, a limited array of antioxidants was tested in varying daily quantities. For instance, beta carotene underwent trials at 30 mg, 50 mg, and 75 mg per day, whereas such comprehensive dosage exploration is lacking for other substances under consideration [26,29,36].

Similar reasoning can also be made for the administration protocols where diverse regimens encompass continuous dosing over 3 years to more abbreviated administrations spanning 3 months [26,29,36]. The duration and frequency of administration, too, could significantly influence the outcomes. Recognizing the intricacies of therapeutic planning becomes tantamount to developing an optimal regimen that could wield a profound impact on outcomes, perhaps rivaling the significance of the molecule itself. Considering these crucial considerations, it becomes evident that labeling a molecule or an entire category, such as antioxidants, as definitively ineffective necessitates a judicious examination. The journey to definitive conclusions remains nuanced and ever-evolving, beckoning further.

Some considerations must also be made on the way in which the second primary neoplasms have been diagnosed. It is imperative when conducting studies of this type to explicitly code the characteristics that distinguish them from a recurrence. Bairati et al. used explanatory criteria starting from the present literature. The characteristics to define it were a distance of at least 2 cm from the primary site, in the case of the same histological type, while instead there were no distance limits for other histological types. In the case of pulmonary manifestations, the nodules had to be single and not have the characteristics of metastasis if they were of the same histotype [38,39]. Unfortunately, the criteria adopted for considering a neoplasm as a second primary cancer were not always reported in the considered literature.

Β-carotene supplementation has been suspected to be related to lung cancer development. In fact, Bairati et al. had to discontinue its administration [27]. Although this evidence was not universally recognized, the ethics committee suggested a revision of the trial protocol [40]. The authors had already administered it to 156 patients that were considered separately in the statistical analysis [26].

The major surprise of their study was the observation that α-tocopherol and β-carotene supplementation was a risk factor for the onset of second cancers during the 3 years of administration (44% increased risk of tumor recurrence, borderline statistical significance), but then, it became a protective factor in the following years. This increased especially when only the first of the two substances were used. This observation leaves us perplexed and waiting for a detailed explanation of the phenomenon [26]. Some authors have hypothesized the role of vitamin supplements in accelerating the growth of latent tumors [41,42].

Continuing smoking after diagnosis and therapy for HNSCC leads to an increased risk of adverse events, disease recurrence, second primary malignancies, and death [7,43,44,45]. Meyer et al. hypothesized that supplementation with α-tocopherol combined with cigarette smoke hinders the action of RT. Tissue hypoxia produced by smoking prevents some of the oxygen-dependent mechanisms of radiation. In addition to being induced to quit smoking, patients should also be discouraged from taking cigarette smoke and α-tocopherol in combination [28]. The negative role of smoking can lead someone to argue that the results of the EUROSCAN trial were derived from a population with a high prevalence of smokers. Even in the light of the evidence of how smoking can influence the action of antioxidants, however, the other trials have not shown any benefits of these drugs on the development of second primitives [28,33].

Seixas-Silva et al. observed only one case of recurrence out of 45 HNSCC patients. They concluded that the bioadjuvant combination used in their phase 2 trial was effective in preventing recurrence and second primary tumors [31]. Considering the absence of a control group and the small number of participants in their trial, we cannot assume their conclusion of the efficacy of this combination of supplements. The same can be stated for the other phase 2 study that investigated the association of interferon-α, 13-cis-retinoic acid, and α-tocopherol [32].

Some authors investigated whether higher blood concentrations of β-carotene were associated with a reduction in the side effects of RT in patients with HNSCC. Higher dietary β-carotene intake and higher plasma levels are associated with fewer serious side effects OR = 0.61 [95% confidence interval (CI) = 0.40–0.93] and OR = 0.73 (95% CI = 0.48–1.11), respectively. In addition, higher plasma levels of beta carotene were also associated with a lower incidence of local recurrences (HR = 0.67; 95% CI = 0.45–0.99), while α-tocopherol had no action [46].

Topical and systemic antioxidants significantly improved the mucositis severity scores in HNSCC patients who underwent radiotherapy except melatonin. However, the quality of evidence was low, and additional large randomized controlled trials are needed to confirm these results [21].

### 4.2. Potential Antioxidant Molecules Not Yet Investigated

Those results allow us to state that among the molecules studied up to now, there are no antioxidants effective in secondary prevention in patients diagnosed with HNSCC. That is not to say there are not any molecules in the future that could prove effective. Many antioxidants show promising results in in vitro studies or in animal models but have not yet been tested in humans. Some substances not yet tested for the prevention of second primary neoplasms after HNSCC are vitamin D and CYP11A1-derived-D3-hydroxyderivatives [47]. D3 can be activated in the form of the traditional active vitamin D 1,25(OH)_2_D3, or through the action of CYP11A, D3 is converted to 20(OH)D3 and 22(OH)D3. The former is the main product of this synthesis and is sometimes subsequently processed by other enzymes belonging to the cytochrome P450 family [48,49,50,51]. The antiproliferative, pro-differentiating, anti-inflammatory, and anticancer action of these molecules seems to be comparable to that of 1,25(OH)_2_D3 with no toxicity even at high doses in animal models [49,52,53]. These D3 hydroxy derivatives are vitamin D receptor agonists and retinoid-related orphan receptor alpha and gamma inverse agonists. The aforesaid receptors are expressed by human squamous cell carcinoma cells, and in in vitro studies, they have shown an antitumor action [47].

Another molecule with antioxidant properties that could have use in the prevention of secondary primary tumors is melatonin, but even in this case there are no trials on humans to date. In human squamous cell carcinoma cell lines, melatonin has a proapoptotic effect that correlates with the production of ROS with a mechanism of action based on the mitochondrial reverse electron transport [54,55]. The action of melatonin on the mitochondria leads to an aerobic metabolism incompatible with the tumor microenvironment [56]. Its association with verteporfin induces ROS-mediated apoptosis in vitro also in HNSCC cancer stem cells, which are considered responsible for the resistance of tumors to chemotherapy. Melatonin also reduces the expression of markers of the epithelial–mesenchymal transition and metastatic migration MMP-2 and -9 through the ERK1/2/FOSL1 pathway [57,58]. Its role in inhibiting epithelial–mesenchymal transition and cancer diffusion is not universally recognized, and some authors have identified a premetastatic action [59]. The association of melatonin and rapamycin, an inhibitor of the Akt mammalian target of rapamycin (mTOR) signaling pathway, has a synergistic effect by reducing the toxicity of healthy cells and reducing the resistance of tumor cells against rapamycin [60]. The action of melatonin on HNSCC is not limited only to ROS or to the synergistic association with other substances but also appears to act at the epigenetic level [61]. A very interesting discovery concerns the inhibitory action on the expression of PD-L1, one of the main immune escape mechanisms of HNSCC, and the target of new immunotherapy based on immune checkpoint inhibitors [62,63].

Neoplasms, including HNSCC, secrete cytokines, hormones, and neurotransmitters that influence the homeostasis of the organism towards a more favorable condition for their growth [64]. If this imbalance persists, an in-depth study of these mechanisms and how to reverse them could hypothetically be effective in the prevention of secondary primary tumors after HNSCC.

Numerous other potentially effective antioxidants have not yet been studied. Therefore, we cannot state that all antioxidants are ineffective in preventing secondary primary tumors in patients with HSCC. However, those studied so far are ineffective.

### 4.3. Study Limitations

The limits of this systematic review mainly concern the paucity of publications on the subject. The literature is full of articles concerning antioxidants in the treatment of head and neck cancer, but their use for prevention of second cancers has been studied on few occasions. Although the conclusions of this article lean towards a lack of efficacy of antioxidants in secondary prevention, we cannot define this research as exhaustive because many substances have not yet been studied for this use.

The studies identified were very different from each other in terms of the substances studied, the protocols, the populations, etc. In our opinion, a meta-analytic study of the results obtained with this review was not feasible due to the heterogeneity of the populations and the substances considered. The lack of statistical verification is certainly to be filled with a meta-analysis, and we believe that this can be performed in the future when several new homogeneous studies on the prevention of secondary neoplasms have addressed the problem.

## 5. Conclusions

Patients should be discouraged from smoking after the diagnosis of HNSCC because continuing leads to an increase in the mortality, risk of recurrence, and incidence of second primary tumors.

None of the antioxidants tested so far (α-tocopherol, β-carotene, JP, Isotretinoin or 13-cis-retinoic acid, interferon α-2a, vitamin E, retinyl palmitate, and N-acetylcysteine) was effective in preventing second primary tumors in patients with HNSCC, and if associated with cigarette smoke, they can even be harmful. Beta carotene reduces adverse events from radiotherapy and local recurrences.

## Figures and Tables

**Figure 1 antioxidants-12-01753-f001:**
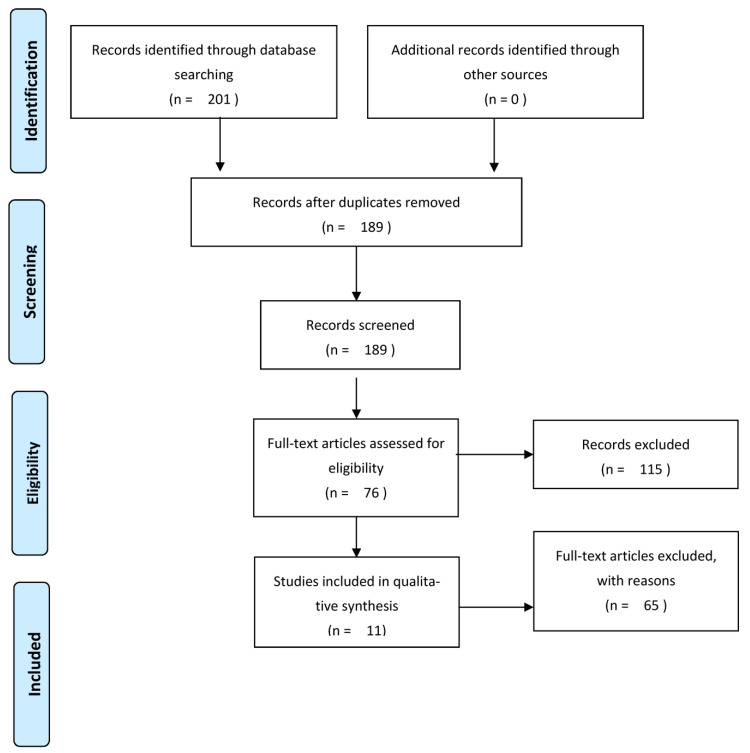
PRISMA flow diagram (see text and Appendix A).

## Data Availability

Not applicable.

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
