# Peer review of "Antioxidant Use after Diagnosis of Head and Neck Squamous Cell Carcinoma (HNSCC): A Systematic Review of Application during Radiotherapy and in Second Primary Cancer Prevention"

_antioxidants, 2023, doi:10.3390/antiox12091753_

Round 1
Reviewer 1 Report
The authors discuss the issue of antioxidants in HNSCC. Based on the review of the literature the authors conclude that antioxidants do not appear effective in preventing second primary tumors in HNSCC patients, and they could only be used in reducing the side effects of radiotherapy. Although conclusion is negative, I see the merit in this study. However, The paper requires revisions.
Description of the materials and methods suggestive restrictive and not full analysis of the problem. I do not know how many papers were missed and why experimental studies were excluded. They can provide proper guidance. Also, number of references for such important problem is rather low.
When they discuss nutritional antioxidants, I am surprised that they do not discuss vitamin D hydroxyderivatives, which show anti-cancer properties against epithelial tumors and anti-oxidative activities (International Journal of Oncology. 2022;61(2). doi: ARTN 9610.3892/ijo.2022.5386; Adv Exp Med Biol. 1268, 257-283, 2020). I am also surprised that melatonin is not mentioned. It has powerful anti-oxidant properties and is non-toxic.
The discussion should be improved. Please also consider that tumors can also produce compounds with anti-oxidative properties of which melatonin is an example, and they could use this capability to escape therapy (How cancer hijacks the body’s homeostasis through the neuroendocrine system, Trends Neurosci 46: 263-275, 2023.,https://doi.org/10.1016/j.tins.2023.01.003). Again this problem deserves proper mentioning.
Discussion remains to be improved and become more inclusive.
Limitations should be provided.
Introduction requires improvements. Not only smoking plays etiological role but also alcohol and unhealthy food.
HPV related category of HNSCC is not mentioned.
Author Response
Dear Reviewers,
We wanted to express sincere gratitude for taking the time to review our article. We genuinely appreciate the effort you've put into reviewing and helping us improve the content.
We carefully considered each of your comments and incorporated all of them to elevate the overall quality of the article.
Changes were made using Microsoft Word's change tracking mode to make changes easily recognizable.
Reviewer 1
Comment 1
“Description of the materials and methods suggestive restrictive and not full analysis of the problem. I do not know how many papers were missed and why experimental studies were excluded. They can provide proper guidance. Also, number of references for such important problem is rather low.”
Reply 1
We updated our materials and methods by adding more details on the manuscript selection process by adding the following sentences:
Line 83-86: “In this phase, all the articles that did not concern trials on human beings were excluded, as well as all the trials that did not have among the outcomes the incidence of secondary primary carcinomas.”
Line 88-92: “The critical issues regarding the analyzed protocols were discussed by our team and summarized in the discussion of this article. Furthermore, the discussion was integrated with a brief dissertation on antioxidant molecules which, although they have known efficacy in primary prevention, have not yet been studied for secondary prevention in patients with HNSCC.”
No experimental studies complying with the requirements of our protocol were excluded.
Regarding the limited number of references, this is due to the paucity of human clinical trials that have investigated the efficacy of antioxidants in preventing secondary primary carcinomas in patients with a previous diagnosis of squamous cell carcinoma of the head and neck. Lastly, thanks to your comments we have increased the references of this article by 79% (from 34 to 61).
Comment 2
“When they discuss nutritional antioxidants, I am surprised that they do not discuss vitamin D hydroxyderivatives, which show anti-cancer properties against epithelial tumors and anti-oxidative activities (International Journal of Oncology. 2022;61(2). doi: ARTN 9610.3892/ijo.2022.5386; Adv Exp Med Biol. 1268, 257-283, 2020). I am also surprised that melatonin is not mentioned. It has powerful anti-oxidant properties and is non-toxic.”
Reply 2
Thank you for this suggestion. We improved our discussion by adding a more precise dissertation about the inefficacy of the antioxidant molecules that have been studied up to now, but it does not exclude that we could find some new effective substances in the future. We added the following sentences:
Line 357-394: “Those results allow us to state that among the molecules studied up to now, there are no antioxidants effective in secondary prevention in patients diagnosed with HNSCC. That is not to say there are not any molecules in the future that could prove effective. Many antioxidants show promising results in in vitro studies or in animal models but have not yet been tested in humans. Between those substances not yet tested for the prevention of second primary neoplasms after HNSCC there are vitamin D and CYP11A1-derived-D3-hydroxyderivatives. [44] D3 can be activated in the form of the traditional active vitamin D 1,25(OH)2D3 or, through the action of CYP11A, D3 is converted to 20(OH)D3 and 22(OH)D3. The former is the main product of this synthesis and is sometimes subsequently processed by other enzymes belonging to the cytochrome P450 family. [45-48] The antiproliferative, pro-differentiating, anti-inflammatory, and anti-cancer action of these molecules seems to be comparable to that of 1,25(OH)2D3 with no toxicity even at high doses in animal models [46,49,50]. These D3 hydroxy derivatives are vitamin D receptor agonists and retinoid-related orphan receptor alpha and gamma inverse agonists. The aforesaid receptors are expressed by human squamous cell carcinoma cells and in in vitro studies have shown an antitumor action [44].
Another molecule with antioxidant properties that could have a use in the prevention of secondary primary tumors is melatonin, but even in this case, there are no trials on humans to date. In human squamous cell carcinoma cell lines melatonin has a pro-apoptotic effect that correlates with the production of ROS with a mechanism of action based on the mitochondrial reverse electron transport [51,52]. The action of melatonin on the mitochondria leads to an aerobic metabolism incompatible with the tumor microenvironment [53]. Its association with verteporfin induces ROS-mediated apoptosis in vitro and also in HNSCC cancer stem cells, which are considered responsible for the resistance of tumors to chemotherapy. Melatonin also reduces the expression of markers of epithelial-mesenchymal transition and metastatic migration MMP-2 and -9 through the ERK1/2/FOSL1 pathway. [54,55]. Its role in inhibiting epithelial-mesenchymal transition and cancer diffusion is not universally recognized and some authors have identified a premetastatic action [56]. The association of melatonin and rapamycin, an inhibitor of the Akt mammalian target of rapamycin (mTOR) signaling pathway, has a synergistic effect by reducing the toxicity of healthy cells and reducing the resistance of tumor cells against rapamycin [57]. The action of melatonin on HNSCC is not limited only to ROS or to the synergistic association with other substances but also appears to act at the epigenetic level [58]. A very interesting discovery concerns the inhibitory action on the expression of PD-L1, one of the main immune escape mechanisms of HNSCC, and the target of new immunotherapy based on immune checkpoint inhibitors. [55,59,60]”
Comment 3
“The discussion should be improved. Please also consider that tumors can also produce compounds with anti-oxidative properties of which melatonin is an example, and they could usethis capability to escape therapy (How cancer hijacks the body’s homeostasis through the neuroendocrine system, Trends Neurosci 46: 263-275, 2023.,https://doi.org/10.1016/j.tins.2023.01.003). Again this problem deserves proper mentioning.”
Reply 3
Thanks for this interesting comment. We didn't hesitate to include it in our discussion, as follows.
Line 395-399: “Neoplasms, including HNSCC, secrete cytokines, hormones, and neurotransmitters that influence the homeostasis of the organism towards a more favorable condition for their growth [61]. Assuming that this imbalance persists, an in-depth study of these mechanisms and how to reverse them could hypothetically be effective also in the prevention of secondary primary tumors after HNSCC.”
Comment 4
“Discussion remains to be improved and become more inclusive”
Reply 4
We improved the structure of our discussion by dividing it into 3 paragraphs (4.1 Results discussion, 4.2 Potential antioxidant molecules not yet investigated, 4.3 Study limitations).
Furthermore, thanks to your valuable observations, we have made it complete and more organized. We also focused on stating that it is not possible to say that all antioxidants are ineffective, but that only those that have been investigated up to now have not demonstrated a protective action against the development of secondary primary neoplasms after HNSCC.
Line 400-402: “Numerous other potentially effective antioxidants have not yet been studied; therefore, we cannot state that all antioxidants are not effective in preventing secondary primary tumors in patients with HSCC, but only those studied so far are. “
Comment 5
“Limitations should be provided.”
Reply 5
We have inserted a paragraph with the limitations of the study at the end of our discussion.
Line 404-416: “The limits of this systematic review mainly concern the paucity of publications on the subject. The literature is full of articles concerning antioxidants in the treatment of head and neck cancer, but their use to prevent second cancers has been studied on a few occasions. Although the conclusions of this article lean towards a lack of efficacy of antioxidants in secondary prevention, we cannot define this research as exhausted because many sub-stances have not yet been studied for this use. The studies identified were very different from each other in terms of substances studied, protocols, populations, etc. In our opinion, a meta-analytic study of the results obtained with this review was not feasible due to the heterogeneity of the populations and substances considered. The lack of a statistical verification is certainly to be filled and we believe that this can be done in the future with the new studies on the prevention of secondary neoplasms that have addressed the problem.”
Comment 6
“Introduction requires improvements. Not only smoking plays etiological role but also alcohol and unhealthy food.”
Reply 6
We improved our introduction and mentioned alcohol and lack of antioxidants as risk factors for HNSCC development as follows:
Line 32-75: 1“Head and neck cancer is a global health challenge, ranking as the sixth most prevalent malignancy worldwide. With an incidence of 650 000 new cases and 350 000 deaths per year. [1]. HNSCC is a challenging condition to treat, with a 5-year survival of about 60% [2]. The emergence of second primary cancers poses a significant obstacle to successful long-term outcomes. Within the first 5 years of treatment, approximately 5 to 20% of HNSCC patients develop a second primary cancer, [3-5] which has been identified as a leading cause of mortality, even in cases of early-stage neoplasms that have been effectively treated [6]. The risk of developing a second primary cancer has been estimated to be 3 to 5% per year from the diagnosis of the primary neoplasm [7]. Epidemiologic evidence linked low dietary intake of antioxidants to an increased risk of cancer, particularly squamous cell carcinoma [8], [9]. Therefore, antioxidants have been studied for their possible use in neoplasm chemoprevention [10]. The primary risk factors for developing HNSCC are cigarette smoking and alcohol consumption. [11] The spread of the human papillomavirus is increasing the incidence of related cervico-cephalic neoplasms. [12] Conversely, a diet rich in antioxidants is a protective factor against the development of head and neck cancer. Patients most at risk, i.e. smokers, are also those who consume the least amount of fruit and vegetables [13-15]. This has led to the hypothesis that integrating antioxidants into the diet could not only play a role in primary prevention but also in secondary prevention for at-risk individuals. Furthermore, some HNSCC patients also take antioxidant supplements during chemotherapy or radiotherapy to alleviate side effects. [16] However, their impact on cancer outcomes remains uncertain [16-19]. Given the protective role of antioxidants against tumors, several researchers have independently wondered whether supplementation in patients with HNSCC could reduce the incidence of second primary neoplasms. The aim of this systematic review of the literature is to summarize the evidence on the subject and provide an answer on the potential action of these molecules in preventing second primary cancers.”
Comment 7
“HPV related category of HNSCC is not mentioned.”
Reply 7
We mentioned the human papillomavirus as a risk factor for the development of the disease, but among the relevant articles for our systematic review, the authors did not take into consideration the possible HPV positivity in the study populations. For this reason, although we agree with you on the importance of this infection in the epidemiology of the disease, the discussion on it has been limited.
Reviewer 2 Report
This report examines the literature and concludes that a case cannot be made for the use of antioxidants for decreasing adverse effects of head & neck cancer. The title suggests that there could be a role for antioxidants, so it needs to be changed. Perhaps words such as ‘ineffective’ or ‘do not’ can be introduced. The title gives the impression that there is a positive effect, as is the statement that begins with ‘a diet rich in antioxidants . . ’ How reliable are these data?
In a literature reviews it cannot necessarily be assumed that everything published must be true. Many examples disputing this claim could be cited. How good is the evidence that antioxidants can reduce adverse effects of ionizing radiation?
In some of the cited examples, it appears that antioxidants made thing worse. The authors appear to be biased in favor of a positive role for antioxidants. An example can be found in lines 131-132: Unexpectedly, the antioxidant group had a statistically significant higher incidence . . What is is ‘unexpected’ about this result. The careful researcher does not ‘expect’, he looks for evidence without any expectations. Lines 195-196 describe a study where the control group of a significant number of patients, had a lower (but not statistically significant) incidence of recurrent cancer. This may have also been unexpected.
The conclusions do not reflect that is contained in the title. Cigarette smoking is not mentioned in the title and does not involve any antioxidants. This is a separate topic although the results are not surprising. Effects of beta-carotene are mentioned in the context of radiotherapy. How does this relate to antioxidants and second primary cancers? The authors need a better title. Perhaps ‘An examination of the role of antioxidants in treatment of head & neck cancer’ or something similar.
How good is the data indicating that smokers tend to consume fewer antioxidants? That diets rich in antioxidants protect from head & neck cancer? Is there any mechanism proposed that might indicate a mechanism whereby antioxidant could either antagonize the effects of carcinogens on mammalian cells? Cancer recurrence means that either the therapy did not eradicate all tumor or additional tumor arose after successful therapy. It seems unlikely that antioxidants will affect efficacy of therapy. The only rationale for use of antioxidants would be that appearance of new tumors would be inhibited. This would imply interference with a carcinogenic process.
Author Response
Dear Reviewers,
We wanted to express sincere gratitude for taking the time to review our article. We genuinely appreciate the effort you've put into reviewing and helping us improve the content.
We carefully considered each of your comments and incorporated all of them to elevate the overall quality of the article.
Changes were made using Microsoft Word's change tracking mode to make changes easily recognizable.
Reviewer 2
Comment 1 and 4
“This report examines the literature and concludes that a case cannot be made for the use of antioxidants for decreasing adverse effects of head & neck cancer. The title suggests that there could be a role for antioxidants, so it needs to be changed. Perhaps words such as ‘ineffective’ or ‘do not’ can be introduced. The title gives the impression that there is a positive effect, as is the statement that begins with ‘a diet rich in antioxidants . . ’ How reliable are these data?”
“The conclusions do not reflect that is contained in the title. Cigarette smoking is not mentioned in the title and does not involve any antioxidants. This is a separate topic although the results are not surprising. Effects of beta-carotene are mentioned in the context of radiotherapy. How does this relate to antioxidants and second primary cancers? The authors need a better title. Perhaps ‘An examination of the role of antioxidants in treatment of head & neck cancer’ or something similar.”
Reply 1 and 4
Thank you for your suggestion, we changed our title considering your observations:
Title: “Antioxidant use after diagnosis of head and neck squamous cell carcinoma (HNSCC): A systematic review of application during radiotherapy and in second primary cancer prevention”
Comment 2
“In a literature reviews it cannot necessarily be assumed that everything published must be true. Many examples disputing this claim could be cited. How good is the evidence that antioxidants can reduce adverse effects of ionizing radiation?”
Reply 2
We agree with you that everything published is not to be acquired uncritically. This is the pillar of the experimental sciences. Regarding the use of antioxidants, we inserted this section into our discussion. We cited a recent systematic review about the subject that critically summarizes all the studies in the literature and highlighted their grade of evidence.
Line 383-386: “Topical and systemic antioxidants significantly improved mucositis severity scores in HNSCC patients who underwent radiotherapy except melatonin. However, the quality of evidence was low and additional large randomized controlled trials are needed to confirm these results [19]”
Comment 3
“In some of the cited examples, it appears that antioxidants made thing worse. The authors appear to be biased in favor of a positive role for antioxidants. An example can be found in lines 131-132: Unexpectedly, the antioxidant group had a statistically significant higher incidence . . What is is ‘unexpected’ about this result. The careful researcher does not ‘expect’, he looks for evidence without any expectations. Lines 195-196 describe a study where the control group of a significant number of patients, had a lower (but not statistically significant) incidence of recurrent cancer. This may have also been unexpected.”
Reply 3
Thanks for pointing this out to us. We fully agree with your comments. We have corrected the quoted sentences as below:
Lines 199-201: “The antioxidant group had a statistically significant higher incidence of second primary cancers than the placebo group (60/1000 person-years vs 25/1000 person-years, hazard ratio (HR) = 2.42 (95% CI = 1.45 to 4.04).”
We replaced the sentence: “An interesting fact that emerged from it was that there is no benefit on the development of second primary neoplasms from this integration, but rather that the fourth group that did not receive any treatment had a lower incidence of secondary neoplasms, even if not statistically significant [28], [29]”
with, lines 263-266: “The authors observed no benefit on the development of second primary neoplasms from this integration; the fourth group that did not receive any treatment had a not statistically significant lower incidence of secondary neoplasms [31,32]”.
Comment 4
“How good is the data indicating that smokers tend to consume fewer antioxidants? That diets rich in antioxidants protect from head & neck cancer? Is there any mechanism proposed that might indicate a mechanism whereby antioxidant could either antagonize the effects of carcinogens on mammalian cells? Cancer recurrence means that either the therapy did not eradicate all tumor or additional tumor arose after successful therapy. It seems unlikely that antioxidants will affect efficacy of therapy. The only rationale for use of antioxidants would be that appearance of new tumors would be inhibited. This would imply interference with a carcinogenic process.”
Reply 4
The data indicating that smokers tend to consume fewer antioxidants derives from the paper of McLure et al. (McClure JB, Divine G, Alexander G, Tolsma D, Rolnick SJ, Stopponi M, Richards J, Johnson CC. A comparison of smokers' and nonsmokers' fruit and vegetable intake and relevant psychosocial factors. Behav Med. 2009 Spring;35(1):14-22. doi: 10.3200/BMED.35.1.14-22. PMID: 19297300; PMCID: PMC2687811) that has been cited in our article. The manuscript describes a trial with a population of 2540 patients in the US. We know how difficult this type of data is to generalize, especially worldwide, but considering the sample size, we preferred to not ignore this observation. Thank you for your observation, to make this statement more reliable to the reader, we have included additional bibliographic elements that have observed this difference in dietary behavior (McPhillips JB, Eaton CB, Gans KM, Derby CA, Lasater TM, McKenney JL, Carleton RA. Dietary differences in smokers and nonsmokers from two southeastern New England communities. J Am Diet Assoc. 1994 Mar;94(3):287-92. doi: 10.1016/0002-8223(94)90370-0. PMID: 8120293; Millen BE, Quatromoni PA, Nam BH, O'Horo CE, Polak JF, Wolf PA, D'Agostino RB; Framingham Nutrition Studies. Dietary patterns, smoking, and subclinical heart disease in women: opportunities for primary prevention from the Framingham Nutrition Studies. J Am Diet Assoc. 2004 Feb;104(2):208-14. doi: 10.1016/j.jada.2003.11.007. PMID: 14760568; Osler M. The food intake of smokers and nonsmokers: the role of partner's smoking behavior. Prev Med. 1998 May-Jun;27(3):438-43. doi: 10.1006/pmed.1998.0289. PMID: 9612834).
Round 2
Reviewer 1 Report
The authors adequately revised the manuscript
Author Response
Dear reviewers,
Thank you for your meticulous input in reviewing our article and helping us improve it. We have followed all the indications you have given us in these second revisions. We thank you for these new comments and for taking the time to help us so carefully.
Again, corrections to the text have been made using Microsoft Word's change-tracking mode to make it easier for you to spot them.
Reviewer 1
Comment 1
“The authors adequately revised the manuscript.”
Reply 1
Thank you for your review, you have helped us to develop a more complete and better-structured article. We are proud to know that we were able to meet your requirements.
Reviewer 2 Report
In this report, the authors conclude that none of the agents they define as ‘antioxidants’ affect what are termed ‘secondary primary tumors’ in patients with of head & neck tumors. The carcinogenic effects of cigarette smoke should be well-known. What is a secondary primary tumor and how is this distinguished from recurrence of the initial (primary) tumor? Assuming that these are really new tumors, it is perhaps not surprising that conditions that led to appearance of one tumor can lead to appearance of another. It is also possible that what is termed a ‘second primary’ was always present.
The new title looks good. Lines 25-26 of the Abstract are confusing. ‘. . none of the antioxidants tested . . do not appear effective . . ‘ A double negative makes no sense. What the authors appear to mean is ‘none of the antioxidants . . appear to be effective . . ‘ It might be useful to define ‘antioxidant’ and point out how such agents might affect the carcinogenic process. In order for ingested antioxidants to have an effect on carcinogenic processes, they need to reach sites where things are happening. It is assumed that these agents somehow survive the digestive processes? Is there evidence for this?
The material relating to Table 1 (lines 134-175) tends to immerse the reader in a vast collection of numbers and other data while not emphasizing the conclusions (if any). It is not necessary to reiterate data shown in the table. What the reader wants to know is whether there is evidence for efficacy of antioxidants. This is true for other summaries of data. The authors get to the point in section 4. (I suggest changing line 241 to just ‘results’. Readers already know that this is a ‘discussion’). It is concluded that antioxidants are mainly ineffective. Since it is not clear exactly what ‘antioxidant’ means this is perhaps not surprising. Lines 294-301 should remind readers of the data, How reliable is the inference that some agents have different effects depending on the timing? What are ‘primitives’ (line 312)?
Aside from the question of what a ‘second primary’ means, the pertinent questions are these: [1] Why would antioxidants be expected to effect the progression of carcinogenesis? [2] How reliable is the evidence that low intake of antioxidants affect the risk of cancer? The authors first conclude that smoking cigarettes leads to cancer, a well-known finding. They then conclude that antioxidants are useless for cancer control. They point out that there are few publications on this topic, that many substances have not been studied, and that more studies are needed. None of this is surprising.
Summary: the presentation of data is needlessly complex, the authors eventually get to the conclusion that there is not enough data to come to a conclusion and that people should stop smoking. A more useful review would begin with a discussion of why use of antioxidants will likely be of no value in preventing tumors from appearing, discuss the limitations of current research and perhaps propose that this line of research is leading nowhere.
Only a few issues.
Author Response
Dear Reviewers,
Thank you for your meticulous input in reviewing our article and helping us improve it. We have followed all the indications you have given us in these second revisions. We thank you for these new comments and for taking the time to help us so carefully.
Again, corrections to the text have been made using Microsoft Word's change-tracking mode to make it easier for you to spot them.
Reviewer 2
Comment 1
“In this report, the authors conclude that none of the agents they define as ‘antioxidants’ affect what are termed ‘secondary primary tumors’ in patients with of head & neck tumors. The carcinogenic effects of cigarette smoke should be well-known. What is a secondary primary tumor and how is this distinguished from recurrence of the initial (primary) tumor? Assuming that these are really new tumors, it is perhaps not surprising that conditions that led to appearance of one tumor can lead to appearance of another. It is also possible that what is termed a ‘second primary’ was always present.”
Reply 1
We agree with you that the definition of a second primary tumor is one of the crucial points of this kind of research. For this reason, we have included the following paragraph in the text which defines it and highlights how only one of the studies considered gives a clear definition of what is a second primary tumor. We believe, as clarified in our manuscript, that future clinical trials should include an objective definition of a second primary tumor.
“Some considerations must also be made on the way in which the second primary neoplasms have been diagnosed. It is imperative when conducting studies of this type to explicitly code the characteristics that distinguish them from a recurrence. Bairati et al., used explanatory criteria starting from the present literature. The characteristics to define it were a distance of at least 2 cm from the primary site, in the case of the same histological type, while instead there were no distance limits for other histological types. In the case of pulmonary manifestations, the nodules had to be single and not have the characteristics of metastasis if they were of the same histotype [35], [36]. Unfortunately, the criteria adopted for considering a neoplasm as a second primary cancer have not always been reported in the considered literature.”
Comment 2a
“The new title looks good.[…]”
Reply 2a
Thanks for the comment on our new title. We have tried to apply all your indications and we are happy to have satisfied them.
Comment 2b
“[…] Lines 25-26 of the Abstract are confusing. ‘. . none of the antioxidants tested . . do not appear effective . . ‘ A double negative makes no sense. What the authors appear to mean is ‘none of the antioxidants . . appear to be effective . .[…]”
Regarding lines 25 and 26, we apologize for the typo. Thanks for pointing that out. We have promptly corrected the sentence as below.
Reply 2b
“In conclusion, none of the antioxidants tested so far (α-tocopherol, β-carotene, JP, Isotretinoin, interferon α-2a, vitamin E, retinyl palmitate, N-acetylcysteine) appear to be effective in preventing second primary tumors in HNSCC patients, and they could only be used in reducing the side effects of radiotherapy.”
Comment 2c
“[…] It might be useful to define ‘antioxidant’ and point out how such agents might affect the carcinogenic process. […]”
Reply 2c
We defined ROS and antioxidants by adding this section to our introduction:
“Pro-oxidant substances or free radicals are molecules equipped with unpaired electrons which, due to their instability, interact with other molecules. Reactive oxygen species (ROS) attack cellular molecules such as fats, proteins, and nucleic acids. [8] Oxidative stress and ROS play causal roles in the development of cancer. ROS damage DNA causing mutations and activating the expression of genes involved in DNA repair and cell proliferation. [9] Antioxidants are molecules able to inhibit oxidation. [8]”
Comment 2d
“[…] In order for ingested antioxidants to have an effect on carcinogenic processes, they need to reach sites where things are happening. It is assumed that these agents somehow survive the digestive processes? Is there evidence for this?”
Reply 2d
We too have asked ourselves this question. As highlighted in the study by Datta et al. of 2018 (DOI: 10.1177/1534735416684947) oral intake of antioxidants is associated with an increase in their circulating levels so we can say that these are absorbed in the intestine and spread throughout the body through the bloodstream.
Comment 3
“The material relating to Table 1 (lines 134-175) tends to immerse the reader in a vast collection of numbers and other data while not emphasizing the conclusions (if any). It is not necessary to reiterate data shown in the table. What the reader wants to know is whether there is evidence for efficacy of antioxidants. This is true for other summaries of data.[…]”
Reply 3
We have removed from this part of the discussion data that is not essential to the description of our outcome. The paragraphs deleted from the text are:
“Patients receiving supplementation experienced 48 of 273-second primary neoplasms within the first 3½ years and 15 of 162 after this period, while the placebo group experienced 21 of 267 and 29 of 193. Patients taking both α -Tocopherol and β -carotene had an incidence of second primary cancer of 10 / 79 vs. 7 / 77 for the placebo at 3½ years and 9 / 57 vs. 12 / 64 for the placebo after that period. Those taking α -Tocopherol alone were 10 / 79 versus 14/190 at 3½ years and 6 / 105 versus 17 / 129 after that time.” And “The authors observed a 42% incidence of side effects in 62 patients (19 had yellowing of the skin, 3 other skin side effects, 3 neurologic symptoms 1 genitourinary, 3 flu-like, 1 endocrine, and 1 cardiovascular) in the experimental group. In the placebo population there 32 patients (16%) referred side effects (6 skin, 2 neurologic, 1 genitourinary, 23 gastrointestinal). During the trial, no grade 4 side effects were observed (life-threatening) and 1 grade 3 (severe) was experienced by a patient in the placebo arm for abdominal pain.”
The text of this part of the results is smoother as below:
“Bairati et al. conducted a multicentric double-blinded placebo-controlled randomized trial comparing supplementation with α-tocopherol (400 IU/day) and β-carotene (30 mg/day) with placebo in patients with stage I or II HNSCC treated with radiotherapy (RT). Antioxidants were administered from the first day of RT and continued for 3 years after its completion. Among 540 patients enrolled, 273 were assigned to the experimental arm and 267 to the control group. The median follow-up was 52 months. The main outcome was to assess whether the supplementation would reduce the incidence of second primary cancers, the second objective was the effect of these substances on RT adverse effects, and the third one was the cancer-free survival impact [26]. β-carotene supplementation was discontinued due to the emerging evidence about its possible association with lung cancer development and the trial was continued with α–tocopherol only [27]. The analysis was also conducted separately for the first 156 patients who used both substances and for the 384 patients subsequently enrolled. The antioxidant group had a statistically significant higher incidence of second primary cancers than the placebo group (60/1000 person-years vs 25/1000 person-years, hazard ratio (HR) = 2.42, 95% CI = 1.45 to 4.04). Especially when α-tocopherol was the only supplement (HR = 2.88, 95% CI = 1.56 to 5.31). After supplements suspension, however, the experimental group had a lower incidence of second neoplasms (39/1000 people per year against 69/1000 people per year, HR = 0.57 95% CI = 0.31 to 1.0), with even greater significance when α -tocopherol was the only supplement (HR = 0.41, 95% CI = 0.16 to 1.03). [26] (Table 1).”
Comment 3a
“[…] The authors get to the point in section 4. (I suggest changing line 241 to just ‘results’. Readers already know that this is a ‘discussion’).”
We have corrected the title of paragraph 4.1 as you indicated.
“[…] It is concluded that antioxidants are mainly ineffective. Since it is not clear exactly what ‘antioxidant’ means this is perhaps not surprising. […]”
Thanks to your observation in comment 1, we have given a definition of an antioxidant in the introduction.
“[…] Lines 294-301 should remind readers of the data, How reliable is the inference that some agents have different effects depending on the timing? […]”
Reply 3a
We have included this observation in our discussion since there are no comparative studies on the different doses. In the previous paragraph, we used expressions such as "leaving the open possibility" precisely to underline the hypothetical nature of the statement. The paragraph you quoted begins with "a similar reasoning" precisely to indicate the same type of statement. Everything is yet to be tested.
Comment 3b
“[…] What are ‘primitives’ (line 312)?”
Reply 3b
We meant “primitive cancers”, we corrected it using those words.
Comment 4
“Aside from the question of what a ‘second primary’ means, the pertinent questions are these: [1] Why would antioxidants be expected to effect the progression of carcinogenesis? [2] How reliable is the evidence that low intake of antioxidants affect the risk of cancer? The authors first conclude that smoking cigarettes leads to cancer, a well-known finding. They then conclude that antioxidants are useless for cancer control. They point out that there are few publications on this topic, that many substances have not been studied, and that more studies are needed. None of this is surprising.
Summary: the presentation of data is needlessly complex, the authors eventually get to the conclusion that there is not enough data to come to a conclusion and that people should stop smoking. A more useful review would begin with a discussion of why use of antioxidants will likely be of no value in preventing tumors from appearing, discuss the limitations of current research and perhaps propose that this line of research is leading nowhere.”
A second primary cancer is a neoplasm that occurs in a patient who has already had a previous diagnosis of cancer and is not attributable to its recurrence or recurrence, or to its metastasis, but rather a new tumor that occurs.
Patients with squamous cell carcinoma of the head and neck are individuals who, in most cases, have been exposed to carcinogenic substances such as smoking and/or alcohol. In this population, the risk factor also remains for the development of new tumors even if the patients are cured of the first one. In fact, the incidence of new cancer in this population is estimated to be up to 20%. It becomes useful to identify, if any, substances that reduce this value. As indicated in the literature and in our introduction, antioxidants are effective in reducing the risk of developing squamous cell carcinoma of the head and neck, as well as numerous other malignancies. Several authors wondered if they were also effective in preventing these new tumors, the second primitive ones. We have carried out this systematic review to answer this question and it has emerged that this line of research has not yet led to an effective substance. From the point of view of cigarette smoking, specifying that patients should not smoke is not a foregone conclusion since these people can take antioxidants to reduce the side effects of radiotherapy, but the combination of these molecules with cigarette smoke has even proved harmful. Our specification that these patients should quit smoking is also related to this new finding.
Regarding the limitations of the study, we have inserted a paragraph about it on your indication in the discussion.
Thanks for the valuable critical insights that helped us improve our article.
Best regards,
Dr. Meliante Piero Giuseppe